# Modified Highland Barley Regulates Lipid Metabolism and Liver Injury in High Fat and Cholesterol Diet ICR Mice

**DOI:** 10.3390/foods11244067

**Published:** 2022-12-16

**Authors:** Jinfeng Zhou, Leiyan Wu

**Affiliations:** 1College of Food Science and Engineering, Jiangxi Agricultural University, Nanchang 330045, China; 2Guangdong JiaBake Food Co., Ltd., A Building, Hengxingchang Industrial Park, Chashan Town, Dongguan 523382, China

**Keywords:** modified highland barley, hyperlipidemia, high fat and cholesterol diet, untargeted metabolomics, gut microbiota

## Abstract

Highland barley (*Hordeum vulgare* L. HB) has been demonstrated to have a series of dietotherapy values, including being low fat, low sugar, high fiber, and especially high in β-glucan. Long-term consumption could reduce the incidence of chronic diseases and metabolic syndromes. In this study, the regulating effect of modified highland barley (MHB) products, namely microwave fluidized HB, extruded and puffed HB, and ultrafine pulverized HB on lipid metabolism and liver injury in mice fed a high fat and cholesterol diet (HFCD) was investigated using microbiota diversity gene sequencing and untargeted metabolomics. A total of six groups of mice were supplemented with a normal diet or an HFCD, with or without MHB, and the experimental period lasted 10 weeks. The obtained results demonstrated that MHB supplementation could effectively reverse the increase in body weight gain and adipose tissue accumulation caused by an HFCD (*p* < 0.05). Moreover, serum biochemical parameters showed that MHB supplementation significantly decreased total cholesterol (TC), triglyceride (TG), and low-density lipoprotein cholesterol (LDL-C) levels, while increasing high-density lipoprotein cholesterol (HDL-C) levels. The results of hematoxylin and eosin (H&E) assays showed that MHB supplementation could significantly improve the liver injury and adipose tissue accumulation. In addition, 16S rRNA amplicon sequencing showed that MHB supplementation increased the bacteroidetes/firmicutes ratio and the abundance Lactobacillus abundance, while also decreasing the Proteobacteria abundance, which are bacteria closely associated with the hyperlipidemia caused by HFCD. LC-MS metabolomics indicated that MHB supplementation significantly enhanced the levels of Deoxycholic acid, Myclobutanil, 3-Epiecdysone, 3,4-Dihydroxybenzeneacetic acid, and so on. In addition, MHB supplementation promoted activation of the Arachidonic acid metabolism pathways, the expression of ABC transporters, bile secretion, primary bile acid biosynthesis, and so on. Above all, this study showed the potential capacity of MHB to relieve hyperlipidemia and provides a reference for developing a new dietary intervention supplement to ameliorate hyperlipidemia.

## 1. Introduction

Obesity is among the most important and significant disease risk factors that threaten the global population. Obesity is induced by increased adipose tissue mass and is characterized by both hyperplasia and hypertrophy [1]. In addition, obesity is closely associated with a large proportion of chronic metabolic diseases. A continuously increased range of obesity is caused by decreased movement, increased food energy intake, and unhealthy lifestyles. Obesity leads to an abnormal glucose and lipid metabolism, which increases the risk of chronic diseases, including hyperlipidemia, hyperglycemia, liver injury, intestinal microbial dysbiosis, and so on [2]. Cereal diet intervention has been confirmed to be a nutritional strategy that can inhabit the development of hyperlipidemia and obesity [3]. Previous research has proven that whole grain highland barley could improve obesity-related diseases, including abnormal lipid metabolism, liver damage, oxidative stress, and so on.

Highland barley (*Hordeum vulgare* L. HB) is the fourth most produced cereal and is mainly distributed in highland areas across the world, including in China. HB is a whole grain and a kind of hull-less barley cultivar [4]. The growing literature on this cultivar has proven that HB is abundant in β-glucan, total polysaccharides, dietary fibers, free phenols, combined phenols, and so on, which are closely related to the health-promoting benefits of HB and its effects on hyperlipidemia, liver damage, and hyperglycemia [5]. Clinical trials indicated that highland barley product consumption significantly decreased body weight and serum glucose, had an inhibitory effect on obesity, and altered the composition of gut microbiota; and in vitro tests indicated that highland barley improved oxidation resistance and slowed glycemic potency [6]. It has been proven that HB can alleviate hyperlipidemia, improve liver injury, decrease adipose tissue accumulation, ameliorate gut microbiota disorders, and regulate metabolism [7].

Previous research proved that whole grain HB can improve gut microbiota and metabolic disorders and further alleviate obesity in mice fed a high-fat diet. These studies showed that gut microbiota shifts were closely associated with the host’s amino acid and lipid metabolism and indicated the potential mechanisms of the hypolipidemic effects of including whole grain HB in the diet [8]. Moreover, the bran of HB has been shown to be the main location of its main active ingredients. Among these bioactive ingredients is β-glucan, which is closely related to hypolipidemia and is proven to reduce cholesterol, decrease triglyceride levels, exert antidiabetic and anticancer activities, and so on [9]. The main ingredients of the outer layer and cell wall of HB are made up of non-starch polysaccharides, which contain hemicellulose, pectin, β-glucan, cellulose, and araboxylan [10]. HB is wrapped in these bioactive ingredients and this protects these cell structures from injury, and it is very difficult to release these chemical ingredients from the cell wall of HB grains. Therefore, in the whole grain food industry, it has become popular to use modern food processing technologies to destroy the outer layer structure and cell wall of whole grains, in order to develop ready-to-eat, barley-based foods with personalized nutritional functions [11]. At present, the main modifying technologies applied to HB are extrusion and puffing, ultrafine grinding, heat fluidization, and microwave fluidization. Research investigating the physicochemical properties (such as antioxidant capacity) of dietary fibers found in HB flour after ultra-fine pulverization has shown that the levels of β-glucan and dietary fibers in HB were higher after ultrafine grinding, which could reduce blood cholesterol and regulate lipid metabolism [12]. Moreover, a previous study showed that after microwaving, baking, and heat fluidization, thermally treated HB possessed a higher reducing power, antioxidant activity, β-glucan extractability, and free phenolics, and lower bound phenolics, total flavonoids, and total phenolics. It was proven that heat fluidization treatment of HB enhanced its nutritional values [13]. However, regardless of the many modification technologies of HB in vitro, there have been few studies about the effects of modified HB products in vivo.

Therefore, this study aimed to investigate the effects of modified highland barley (MHB) (e.g., microwave fluidization, extrusion and puffing, and ultrafine grinding) on hypolipidemia, liver damage, intestinal flora, metabolites, and metabolic pathways in ICR mice fed a high fat and cholesterol diet (HFCD). This study aimed to provide some reference regarding the underlying mechanisms of MHB in hyperlipidemia and to promote the comprehensive utilization of MHB.

## 2. Materials and Methods

### 2.1. Materials

The 3 MHB products, namely microwave fluidized HB, extruded and puffed HB, and ultrafine pulverized HB, were obtained from Qinghai Engineering Technology Research Institute (Xining, China) and named HB-I, HB-II, and HB-III, respectively.

### 2.2. Determination of HB, HB-I, HB-II, and HB-III Components

The dietary fibers, total polysaccharides, free phenols, bound phenols, β-glucan, total anthocyanins, total starch, ash, protein, and fat of HB, HB-I, HB-II, and HB-III were determined based on the methods and standards shown in Table 1.

### 2.3. Animal Experiments and Design

The Jiangnan University Committee on Ethical Management and Use of Laboratory Animals approved this animal experiment (Approval Code: JN. No 20210615c1000). A specific pathogen-free (SPF)-grade animal barrier room was provided to house all the mice at the Experimental Animal Center of Jiangnan University (license number: SYXK (Su) 2021-0056). Sixty ICR mice in total (male, 25.0 ± 3.0 g, 6-week-old) were provided by Jiangsu Jicui Yaokang Biotechnology Co., Ltd. (Nanjing, China). An environment including a room temperature at 21~24 °C, humidity at 50–60%, and a 12 h light/dark cycle was supplied. The mice were given drinking water and a daily normal diet was given freely.

The arrangement of the 6 groups was as follows: normal control group (NCG, fed with AIN-93M diet), high fat and cholesterol group (FG, XT108C diet), highland barley powder group (HBG, XT108C HFCD mixed highland barley), microwave fluidized highland barley HFCD group, HB-1 (fed with XT108C mixed HB-I diet), extruded and puffed highland barley HFCD group, HB-2 (fed with XT108C mixed HB-II diet), and ultrafine pulverized highland barley HFCD group, HB-3 (fed with XT108C mixed HB-III diet).

The XT108C diet, XT108C mixed HB diet, XT108C mixed HB-1 diet, XT108C mixed HB-2 diet, XT108C mixed HB-3 diet, and AIN-93M diet were manufactured and purchased from synergy medicine bioengineering Co., Ltd. of Jiangsu (Nanjing, Jiangsu, China). During the first week pre-feeding period, all 6 groups were fed the AIN-93M diet. After that, FG was supplemented with XT108C HFCD. HBG, HB-1, HB-2, and HB-3 were fed an XT108C mixed HB diet, XT108C mixed HB-1 diet, XT108C mixed HB-2 diet, and XT108C mixed HB-3 diet, respectively. The XT108C mixed HFCDs were manufactured according to composition levels of total starch, fat, and protein of HB, HB-1, HB-2, and HB-3. Detailed diet compositions are listed in Table 2.

In addition, as can be seen Figure 1, after the first week pre-feeding period, the mice were recorded for their feed intake (FI), body weight (BW), and body weight gain (BWG). On the day of the animal study, all mice were fasted for 8 h. Then, mice were anesthetized and the required tissues, contents, and serum were collected for further analysis.

### 2.4. Investigation of Serum Biochemical Indices Investigation

At the end of the animal experiment day, all the mice were anesthetized and serum was obtained from the orbital vein and then centrifuged. The serum total cholesterol (TC), low density lipoprotein cholesterol (LDL-C), triglyceride (TG), alanine aminotransferase (ALT), high-density lipoprotein cholesterol (HDL-C), and aspartate aminotransferase (AST) levels were determined using a Mindray BS420 Automatic Biochemical analyzer (Shenzhen Mindray, Wuhan Shengshida Medical Equipment Co., Ltd., Wuhan, China) [16].

### 2.5. Histopathology Analysis of Liver and Adipose Tissue

Paraformaldehyde (4%) was used to embed the mice liver and adipose samples. Then, hematoxylin and eosin (H&E) were used to stain the tissue paraffin block. Finally, light microscopy was used to analyze the stained sections at 200 magnification (100 μm) for further observation [17].

### 2.6. Gut Microbiome Assessment

Genomic DNA from mouse feces was obtained using a DNA Kit (Norcross, OMEGA Soil, Omega Bio-Tek, D5625-01, GA, Shanghai, China) and kept at −20 °C for further determination. Shanghai Parsono Biotechnology Co., Ltd. (Shanghai, China) was commissioned to perform the sequencing analysis on the V3–V4 region. PCR amplification was performed using reverse primer (R): GGACTACHVGGGTWTCTAAT and forward primer (F): ACTCCTACGGGAGGCAGCA. The PCR components were as follows: 1 μL (10 uM) of each forward and reverse primer, 5 μL buffer (5×), 2 μL (2.5 mM) dNTPs, 1 μL DNA Template, 0.25 μL Fast pfu DNA Polymerase (5 U/μL), and 14.75 μL ddH2O. Starting denaturation was performed at 98 °C for 5 min. A total of 25 cycles were performed. Denaturation was performed for 30 s. Annealing was conducted for 30 s at 53 °C. Extension was performed for 45 s at 72 °C. Final extension was performed at 72 °C for 5 min. DNA Clean Beads (Vazyme VAHTSTM) were used to purify the PCR amplicons, and an Assay Kit (Invitrogen, Quant-iT PicoGreen dsDNA, Carlsbad, CA, USA) was used to quantify these amplicons and gather them in equal amounts. Paired-end sequencing with a MiSeq Reagent Kit v3 was performed using the Illumina MiSeq platform at Shanghai Personal Biotechnology Co., Ltd. (Shanghai, China) [18].

After the original data were spliced and filtered, the species richness and uniformity information within NCG, FG, HBG, HB-1, HB-2, and HB-3, as well as unique and common operational taxonomic unit (out) data among the 5 groups, were analyzed using species classification analysis and operational taxonomic unit (OUT) clustering. The community structure among NCG, FG, HBG, HB-1, HB-2, and HB-3 was analyzed using principal co-ordinates analysis (PCoA), non-metric multidimensional scaling (NMDS), and other dimensionality reduction methods. The community structure and species composition significances were tested using LefSe, Venn, and other statistical analysis methods.

### 2.7. Nontargeted Metabolomics

First, 2-chlorophenylalanine methanol (−20 °C, 4 ppm) was mixed with mice feces, then ground, extracted via sonication, and centrifuged. The obtained solution was filtered. Liquid chromatography-mass spectrometry (LC-MS) was used to analyze it. The chromatographic indices were as follows: 40 °C column (Waters, ACQUITY UPLC^®^ HSS T3, 1.8 µm, 150 × 2.1 mm); 8 °C autosampler temperature; 5 mM ammonium formate in acetonitrile (B) and water (A) or analytes 0.1% formic acid in water (C); and 0.1% formic acid in acetonitrile (D) gradient elution at a 0.25 mL/min flow rate, with 2 μL per injection. The mass spectrometry indices were as follows: a spray voltage of 3.5 kV and −2.5 kV of ESI-MSn experiments in positive and negative modes; 30 and 10 arbitrary units of sheath gas and auxiliary gas, respectively; a capillary temperature at 325 °C; analyzer scanned *m*/*z* 81–1000, with a 60,000 mass resolution for full scan; a data-dependent acquisition (DDA) study (MS/MS) HCD scan; and a normalized collision energy at 30 eV [19].

All mass spectrometry files (.WIFF) were analyzed using SCIEX OS V1.4, (V1.4, SCIEX, UAS)and chromatographic peaks were integrated and corrected. Each chromatographic peak area (Area) represents the relative quantitative value of its counterpart, and all chromatographic peak area integral data were derived.

### 2.8. Statistical Analysis

QC sample data were obtained to ensure the accuracy and reliability of the data results. Moreover, multivariate statistical analyses, including principal component analysis (PCA) and partial least squares discriminant analysis (PLS-DA), were conducted to reveal the differences in metabolic patterns among the different groups. Hierarchical clustering (HCA) and metabolite correlation analysis were used to reveal the relationships among samples and between metabolites. Finally, the biological significance of the metabolites was explained using a functional analysis of metabolic pathways.

All data were visualized as means ± SD. Statistical differences were detected using one-way ANOVA, followed by the Duncan’s multiple range test using SPSS (Version 20.0, SPSS Inc., Chicago, IL, USA). Values with different letters are significantly different at *p* < 0.05.

## 3. Results

### 3.1. Compositions Analysis of HB, HB-1, HB-2, and HB-3

A total of 10 composition levels of HB, HB-1, HB-2, and HB-3 are visualized in Figure 2. The levels of dietary fibers, β-glucan, total polysaccharides, free phenols, and bound phenols in HB-1 were significantly higher than those in HB, HB-2, and HB-3, while the total anthocyanins, ash, protein, and fat levels of HB-3 were significantly higher than those in HB, HB-1, and HB-2. In addition, the total starch content of HB was higher than that of HB-3, HB-2, and HB-1. The moisture level of HB-2 was greater than that in HB, HB-1, and HB-2.

### 3.2. Body Weight (BW), Body Weight Gain (BWG) and Feed Intake (FI)

The effects of HBG, HB-1, HB-2, and HB-3 on the BW, BWG, and FI in HFCD-fed mice are listed in Table 3. Compared with NCG, the BW and BWG of HB, HB-1, HB-2, HB-3, and FG were significantly enhanced, which demonstrated that the HFCD successfully caused obesity in the mouse model. However, the BW and BWG of HB, HB-1, HB-2, and HB-3 were significantly smaller than FG, while those of HB-1 were significantly higher than those of HB, HB-2, and HB-3, indicating that HB-3 especially attenuated the hyperlipidemia in HFCD-fed mice.

### 3.3. Serum Lipid Parameters

The serum lipid profiles of the HFCD-fed mice supplemented with HB, HB-1, HB-2, and HB-3 are listed in Table 4. The LDL-C, AST, TC, ALT, and TG levels of FG increased, while the HDL-C level was reduced compared to NCG (*p* < 0.05). However, after supplementation with HB, HB-1, HB-2, and HB-3, the TC, TG, LDL-C, AST, and ALT levels were significantly reduced, while the HDL-C levels increased, indicating that HB-3 especially improved the hyperlipidemia and liver injury in the HFCD mouse model.

### 3.4. Liver Damage Analysis

The liver damage sections among HB, HB-1, HB-2, HB-3, NCG, and FG are presented in Figure 3. The H&E section diagrams indicate that the liver tissue cells in NCG display a normal cellular morphology and liver cell structure, with no pathological fat changes. However, FG presented a destroyed liver structure, obscured cellular boundaries, and numerous cytoplasmic vacuolations and lipid droplets in the liver cells. After the 10 weeks of intervention with HB, HB-1, HB-2, and HB-3, the high fat and cholesterol diet induced liver injuries in mice were improved, showing less vacuolization, fewer lipid droplets, and a normal appearance. These results indicate that MHB, especially HB-2, could improve liver damage and provide a vital hepatoprotective effect.

### 3.5. Adipose Histopathology Assessment

As visualized in Figure 3, the HFCD induced increases in the size of adipocytes in the FG compared with the NCG. However, after supplementation with HB, HB-1, HB-2, and HB-3, the size of the adipocytes decreased, and their structures became sound. These diagrams prove that compared with HB, HB-2, and HB-3, HB-1 could significantly ameliorate the increase in adipocyte size and alleviate the structural disorders in these cells caused by the HFCD.

### 3.6. Gut Microbiota Assessment

A total of 30 collected group samples, in six groups (*n* = 5), were investigated in this study. Alpha diversity parameters (Figure 4A) proved that HFCD significantly decreased the Chao1, Faith’s PD, observed species, Pielou’s evenness, and Simpson and Shannon values, while HB, HB-1, HB-2, and HB-3 significantly increased them, which showed that HB, especially HB-1, could reverse the decrease in community diversity, richness, and evenness. Moreover, Good’s coverage parameters indicated that all the groups’ coverages were over 98.5%, indicating that all the selected analysis data were reliable and valid. As shown in Figure 4B–E, the Chao1, Faith’s PD, Good’s coverage, observed species, Pielou’s evenness, and Shannon and Simpson rarefaction curves showed that the curve tended to be flat with the increase in leveling depth, which indicated that the sequencing results adequately reflected the diversity of the current sample of groups with an increase in sequencing depth. In addition, the abundance of intestinal tract species was not increased with an increase in the number of sequenced samples, showing that this study sufficiently reflects the flora characteristics of the samples.

The β diversity can reflect species composition differences on a spatial scale and can thus be used to analyze the similarity among different groups. Principal coordinates analysis (PCoA) and non-metric multidimensional scaling (NMDS) were used to visually exhibit the differences and similarities of the microorganisms among different groups. In Figure 4G–J, the intestinal microbial structures in the NCG and FG were different, and those among HB, HB-1, HB-2, and HB-3 were similar. This showed that HB, especially HB-1, supplementation markedly changed the β diversity of the intestinal microbes. The OUT rank curve (Figure 4F) indicated that HB, HB-2, and especially HB-1 and HB-3 enhanced the evenness of the microbiota community compositions. Consistently with previous studies, highland barley changed the structure of the gut microbiota [20].

In Figure 4K, the average OTU amounts among NCG, FG, HBG, HB-1, HB-2, and HB-3 were 4060, 861, 1003, 2069, 797, and 1224, respectively, with an OTU sequence similarity level of >97%. Among these, 3797, 740, 1806, 534, and 961 were the distinct microbes of NCG, HB-1, and HB-3, while only 598, 740, and 534 were in FG, HBG, and HB-2, which showed that HB, especially HB-1, could clearly recover the amount of operational taxonomic units (OTUs) compared with FG. As shown in Figure 5A, Firmicutes, Bacteroidetes, Proteobacteria, Actinobacteria, Verrucomicrobia, and TM7 were significantly recorded at the phylum level among the six groups. Among these, Firmicutes and Bacteroidetes were the predominant bacteria and occupied a >70% relative abundance across the bacterial communities. In addition, FG significantly decreased the Bacteroidetes/Firmicutes (B/F) ratio compared with NCG HB, and HB-1, HB-2, and HB-3 reversed the decrease in the B/F ratio (*p* < 0.05). Moreover, the abundances of Firmicutes, Deferribacteres, and TM7 in the HB, HB-1, HB-2, and HB-3 groups increased compared with FG, while those of Bacteroidetes, Proteobacteria, Actinobacteria, and Verrucomicrobia decreased (*p* < 0.05). In Figure 5B, Acinetobacter, Bifidobacterium, Mucispirillum, Lactobacillus, Akkermansia, Oscillospira, and Shigella were significantly recorded at the genus level among the six groups. HFCD caused a decrease in the abundance of Lactobacillus, Mucispirillum, Acinetobacter, and Oscillospira, as well as Bifidobacterium, Akkermansia, and Shigella, compared with the NCG. However, HBG, HB-2, and especially HB-1 and HB-3 supplementation increased the relative abundances of Lactobacillus, Mucispirillum, Acinetobacter, and Oscillospira compared with FG. These results are supported by the findings of Zang et al. [20,21]. HB not only upregulated the abundance of beneficial bacteria, but also inhibited the growth of pathogenic bacteria.

LEfSe analysis was used to analyze the microbial community, in order to record specific gut microbiota taxa from phylum to genus, which was related to different dietary interventions that caused remarkable differences. In Figure 5C,D, 21 abundant differential taxa, including four genera among six groups, were identified. At the genus level, Alispites and Anaeroplasma were predominant in the NCG. In contrast, Bifidobacterium and Parabacteroides were the featured genera in the FG. Therefore, the pathogenesis of HFCD-caused disorders might be attributed to changes in these bacteria. Corynebacterium, Adlercreutzia, and Streptococcus were dominant in HB-2, which indicates that they might be potential biomarkers for MHB supplemented mouse intestinal indicators, to reverse HFCD-caused gut microbiota disorders.

### 3.7. Nontargeted Metabolomics Assessment

Nontargeted metabolomics was used to record the specific changes and fecal metabolite differences after MHB supplementation. OPLS-DA images proved that the metabolite profiles were distantly located, regardless of MHB supplementation, in ESI+ or ESI- among NCG and FG, HBG and FG, HB-1 and FG, HB-2 and FG, and HB-3 and FG, proving the stabilizing effect of HB-2, HB, and especially HB-3 and HB-1 on the intestinal metabolism (Figure 6A,B). Moreover, the alterations in metabolites were divergent among NCG and FG, HBG and FG, HB-1 and FG, HB-2 and FG, and HB-3 and FG. VIP value (≥1), statistical tests (*p* < 0.05) and fold change ≥1.5 were used to assess and screen for potential biomarkers. Among NCG and FG, HBG and FG, HB-1 and FG, HB-2 and FG, and HB-3 and FG, 50 different metabolites in total were found, as shown in Table 5. Important changes in NCG and FG were identified in Deoxycholic acid, Myclobutanil, 4-Hydroxyestradiol, Dicyclomine, and 2-Phenylacetamide. Important changes in HBG and FG were identified in Deoxycholic acid, Myclobutanil, 3-Hydroxyflavone, Secoisolariciresinol, and 1,7-Dimethyluric acid. Important changes in HB-1 and FG were identified in Deoxycholic acid, Myclobutanil, 3-Epiecdysone, 2-Phenylacetamide, and Fluvoxamine. Important changes in HB-2 and FG were identified in Myclobutanil, 3-Hydroxyflavone, 4-Hydroxyproline, 4-Hydroxyestradiol, and Hydrogen phosphate. Important changes in HB-3 and FG were identified in Deoxycholic acid, Myclobutanil, 3,4-Dihydroxybenzeneacetic acid, 2-Phenylacetamide, and Lipoxin B4.

A KEGG pathway enrichment analysis was performed on the differential metabolites with known KEGG IDs, in order to identify pathways affected by MHB supplementation. As can be seen in Figure 7A–E and Table 6, changes in NCG and FG were mainly seen in the Arachidonic acid metabolism, ABC transporters, the biosynthesis of steroid hormones, the Arginine and proline metabolisms, and the Tryptophan metabolism. Changes in HBG and FG were mainly seen in the expression of ABC transporters, the Galactose metabolism, the biosynthesis of Steroid hormones, the Phenylalanine metabolism, and the Pyrimidine metabolism. Changes in HB-1 and FG were mainly seen in the Arachidonic acid metabolism, ABC transporters, Serotonergic synapse, bile secretion, and primary bile acid biosynthesis. Changes in HB-2 and FG were mainly seen in ABC transporters, the Tyrosine metabolism, the Tryptophan metabolism, the Nicotinate and nicotinamide metabolism, and Steroid hormones. Changes in HB-3 and FG were mainly seen in the Arachidonic acid metabolism, ABC transporters, the Tyrosine metabolism, the biosynthesis of Steroid hormones, and Neuroactive ligand–receptor interactions. These main KEGG pathways were important in the regulation of hyperlipidemia [6]. In addition, Figure 8 shows the related corrections in the gut microbial composition based on genus level and significantly differential metabolites. Lactobacillus and Adlercreutzia were significant genera that might serve as potential biomarkers for MHB supplemented mice and become intestinal indicators to reverse HFCD-caused gut microbiota disorders [21,22,23]. Lactobacillus was positively related to 6-Hydroxyhexanoic acid, Cytosine, 9,10-Epoxyoctadecenoic acid, and Gulonic acid (*p* < 0.05); Adlercreutzia was positively related to L-Aspartic acid and L-Glutamine and negatively related to Mesaconate, Oxoglutaric acid, and 3-Dehydrosphinganine.

## 4. Discussion

HB is rich and abundant in β-glucan, dietary fibers, protein, phenolic ingredients, and amino acids, and these compounds serve important roles in the hypolipidemic function of HB [13,19,24]. However, the effects and mechanisms of MHB on the regulation of hyperlipidemia, hepatic protection, intestinal flora disorders, and fecal metabolites induced by HFCD remain to be studied. This study proved that hyperlipidemia was successfully induced in mice by 10 weeks of an HFCD. In addition, the HFCD significantly increased all mice’s BW and BWG, but the FI had no distinct difference compared with NCG. Thus, these results show that the differences in FI did not result in weight gain in the studied mice.

TC enhancement induces disordered arterial endothelial cell function and the occurrence of cerebrovascular disorders, including cardiopathy and atherosclerosis [25]. TG increases can lead to an increase in blood viscosity, thrombosis development, and the occurrence of atherosclerosis [26]. LDL-C can cause atherosclerosis if excessively enriched in vascular cells, which are important carriers of plasma sterols. Excess cholesterol can be transported by HDL-C, from different organs to the liver [27]. In this study, FG enhanced the levels of LDL-C, TC, ALT, TG, and AST, while reducing the HDL-C level. After supplementation with HBG, HB-2, HB-3, and especially HB-1, the LDL-C, TC, ALT, TG, and AST levels were significantly reduced, while the HDL-C level increased (*p* < 0.05). H&E diagrams showed that liver damage was alleviated, fat cells were smaller, and the fat accumulation reduced, indicating that HB-1 has the potential capacity to reduce total lipids and ameliorate liver damage. The results are in agreement with a previous study where bioactive ingredients distributed in buckwheat could decrease abnormal BW gain and serum TC, TG, and LDL-C levels and where this effect was attributed to the protective influence of buckwheat’s bioactive compounds on cholesterol [28]. The hyperlipidemia-alleviating effects of MHBs, especially HB-1, are a result of its rich and abundant dietary fibers, free phenols, polysaccharides, β-glucan, anthocyanins, and bound phenols.

Excessive HFCD intake can result in body metabolism disorders (such as liver inflammation, lipid metabolism, and gut microbiota disorders) [29]. The intestinal flora is very important in maintaining health status and in the daily diet.

With the growing interest in the crucial role of intestinal flora in hyperlipidemia, the use of nutritional supplements to alleviate hyperlipidemia and modulate the gut’s microbial composition has received much attention in recent years [30]. This research showed that the PCoA and NMDS diagrams for NCG, FG, and the four HB groups were obviously separated. HB supplementation led to closer clustering with NCG mice, and the four HB groups were distinctly separated from FG. These results show that HB-1 could restore the structure in gut microbiota community disorders [31]. Moreover, MHB reversed the decreasing trend in gut microbial diversity decrease caused by the HFCD. At the phylum level, MHB, and especially HB-1, enhanced the abundance of Deferribacteres, Firmicutes, and Bacteroidetes, while reducing that of Actinobacteria, Verrucomicrobia, and Proteobacteria. In addition, MHB, and especially HB-1, increased the Bacteroidetes/Firmicutes ratio as well. There is a large variety of pathogens, including Helicobacter, Escherichia, Salmonella, and Vibrio spp., in Proteobacteria [32]. Bacteroides and Firmicutes are closely associated with most of the genes related to hyperlipidemia, and previous research has shown that the B/F ratio is very low in the intestinal flora of high fat diet (HFD)-fed groups [33]. These results clearly indicate that MHB, and especially HB-1, modulates the gut microbiota and further ameliorates hyperlipidemia. Furthermore, at the genus level, MHB significantly decreased the abundance of Bifidobacterium and Akkermansia, while increasing the abundance of Mucispirillum, Oscillospira, Acinetobacter, and Lactobacillus. Lactobacillus can produce a variety of short-chain fatty acids (SCFAs) that can restore microflora composition and improve gut health, according to [34]. SCFAs appear to be the important microbial metabolites controlling gut microbiota relationships, and a diverse gut microbiota composition may be a potential treatment for obesity [35]. Some studies have suggested that the increase SCFA might be related to the conversion of amino acids [36]. Above all, these phenomena prove that MHB, especially HB-1, increased the Bacteroidetes/Firmicutes ratio and modulated the abundances of other specific microbiota, to regulate lipid metabolism, exhibiting the capacity to ameliorate hypolipidemia.

Moreover, LC-MS-based metabolomics was used to explore the effects of MHB on the metabolic parameters. In this study, fecal metabolomics showed that MHB significantly induced alterations in gut metabolites. Important changes in the NCG and FG were found in Deoxycholic acid, Myclobutanil, and 4-Hydroxyestradiol. Changes in the HBG and FG were mainly visualized in Deoxycholic acid, Myclobutanil, and 3-Hydroxyflavone. Changes in HB-2 and FG were mainly visualized in Myclobutanil, 3-Hydroxyflavone, and 4-Hydroxyproline. Changes in HB-3 and FG were mainly visualized in Deoxycholic acid, Myclobutanil, and 3,4-Dihydroxybenzeneacetic acid. Significantly and differentially enriched metabolic pathways of NCG and FG were mainly seen in the Arachidonic acid metabolism and ABC transporters. The main metabolic pathways of HBG and FG were ABC transporters and the Galactose metabolism. The main metabolic pathways of HB-1 and FG were the Arachidonic acid metabolism and ABC transporters. The main metabolic pathways of HB-2 and FG were ABC transporters and the Tyrosine metabolism. The main metabolic pathways of HB-3 and FG were the Arachidonic acid metabolism and ABC transporters.

Deoxycholic acid was the most significant metabolite biomarker. It is a kind a bile acid deficient in hydroxyl on C-7 and originates from the loss of an oxygen atom of bile acid. A long-term high fat and cholesterol diet (HFCD) led to increased deoxycholic acid [37]. Taurine and glycine are its main forms in bile. It has been proven that deoxycholic acid has the potential to ameliorate the insulin resistance and regulate lipid metabolism of type 2 diabetes [38,39]. In this study, compared with FG, deoxycholic acid was largely enriched by MHB, especially HB-1, indicating its potential for ameliorating hyperlipidemia by secreting a large amount of deoxycholic acid, which is in agreement with previous research [40]. Moreover, the arachidonic acid metabolism, bile secretion, linoleic acid metabolism, and biosynthesis of unsaturated fatty acids were the most enriched metabolic pathways and were important in ameliorating the lipid metabolism induced by a high-fat diet in hyperlipidemic models [41]. Arachidonic acid, a kind of fatty acid, plays a crucial role in allergies, inflammation, and other organ functions and reactions [42]. Severity of inflammation can be reflected in the arachidonic acid metabolism to some extent [43].

Above all, MHB, and especially HB-1, could significantly ameliorate hyperlipidemia and reduce inflammation, through enriching a series of feces metabolites and activating metabolic pathways closely related to hyperlipidemia, as identified through LC-MS metabonomics.

## Figures and Tables

**Figure 1 foods-11-04067-f001:**
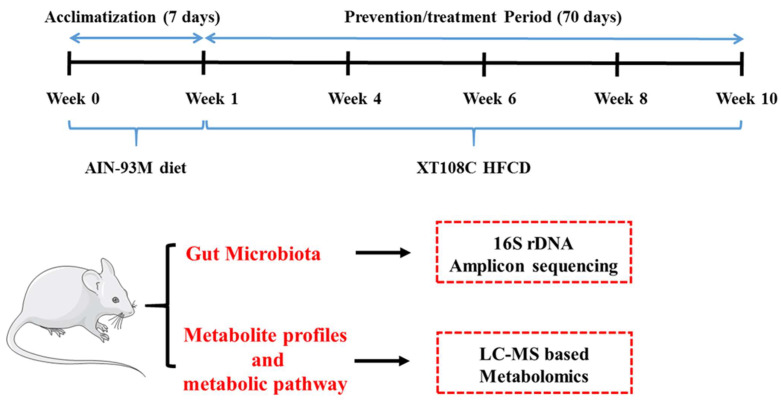
Animal study design. Normal control group (NCG, fed with AIN-93M diet), HFCD group (FG, fed with XT108C diet), HB HFCD group (HBG, fed with XT108C mixed HB diet), HB-1 HFCD group (HB-1, fed with XT108C mixed HB-1 diet), HB-2 HFCD group (HB-2, fed with XT108C mixed HB-2 diet), and HB-3 HFCD group (HB-3, fed with XT108C mixed HB-3 diet).

**Figure 2 foods-11-04067-f002:**
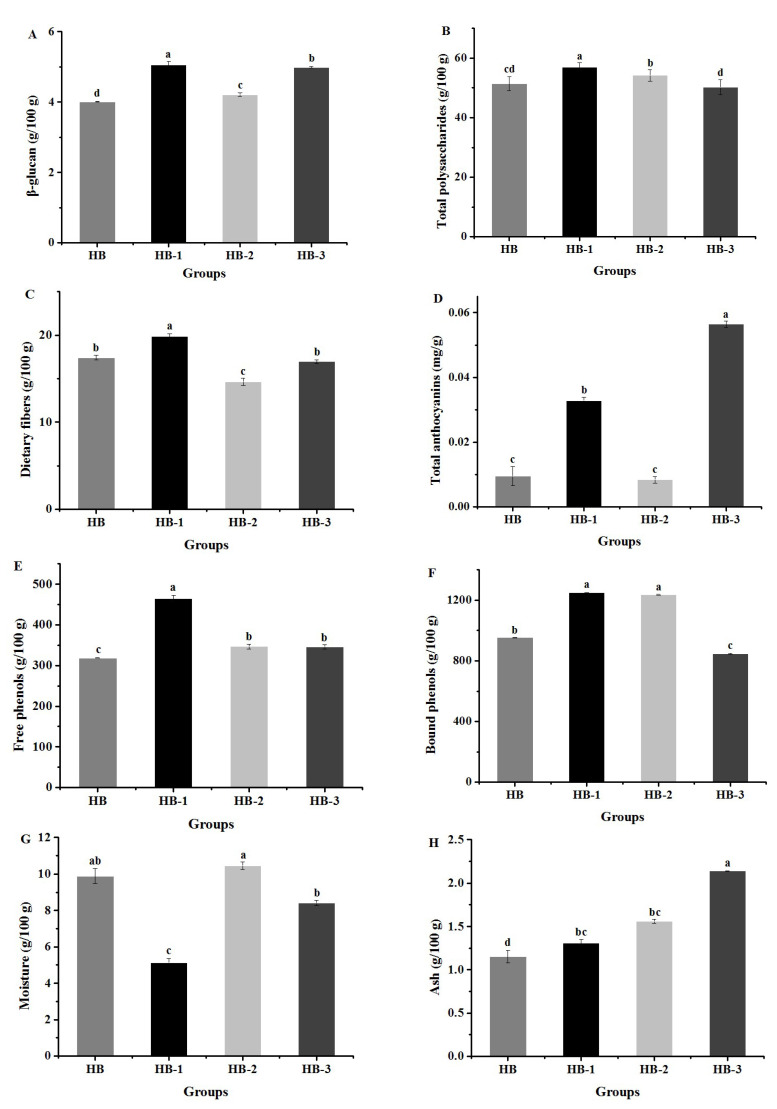
Composition levels of HB (Highland barley), HB-1 (microwave fluidized Highland barley), HB-2 (extruded and puffed Highland barley), and HB-3 (ultrafine pulverized Highland barley).β-glucan (**A**); Total polysaccharides (**B**); Dietary fibers (**C**); Total anthocyanins (**D**); Free phenols (**E**); Bound phenols (**F**); Moisture (**G**); Ash (**H**); Protein (**I**); Fat (**J**); Total starch (**K**). All data are shown as the mean ± SD. Values with different letters are signifi-cantly different at *p* < 0.05.

**Figure 3 foods-11-04067-f003:**
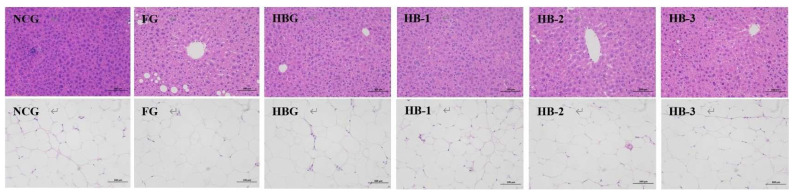
Hematoxylin and eosin (H&E) staining sections of liver and adipose tissue (100 μm). The first row of pictures represent liver tissue; the second row of pictures represent adipose tissue.

**Figure 4 foods-11-04067-f004:**
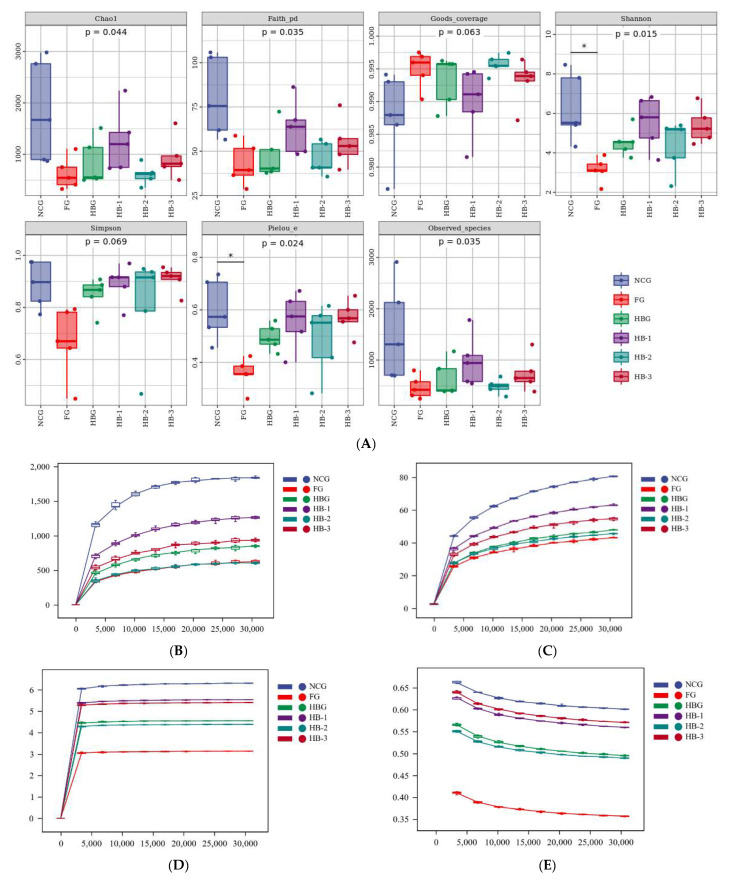
Effect of the six groups on the feces microbiota in HFCD mice. Alpha diversity indexes (**A**), Chao1 rarefaction curve (**B**), Faith’s PD rarefaction curve (**C**), Shannon rarefaction curve (**D**), Pielou’s evenness rarefaction curve (**E**), Rank abundance curve (**F**), PCoA based on bray_curtis distance (**G**), PCoA based on weighted-unifrac distance (**H**), NMDS based on bray_curtis distance (**I**), NMDS based on weighted-unifrac distance (**J**), A Venn diagram showing the overlap of the ASVs (**K**). * indicates a significant difference at *p* < 0.05.

**Figure 5 foods-11-04067-f005:**
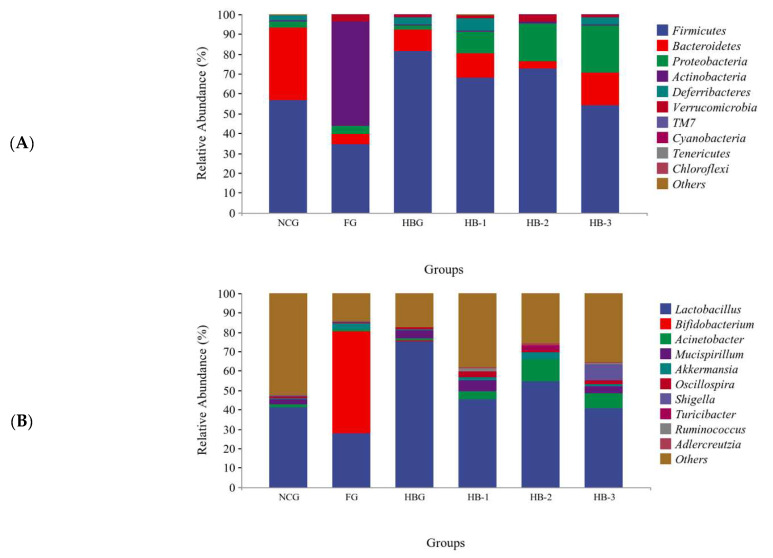
Effect of HB (Highland barley), HB-1 (microwave fluidized Highland barley), HB-2 (extruded and puffed Highland barley), and HB-3 (ultrafine pulverized Highland barley) on feces microbiota in HFCD mice, using linear discriminant analysis effect size (LEfSe) and linear discriminant analysis (LDA) statistical analysis methods. Feces microbial composition at the phylum level (**A**). Feces microbial composition at the genus level (**B**). Cladogram visualizing the output of the LEfSe analysis (**C**). The most significant difference of gut microbial taxa among groups after LDA (**D**).

**Figure 6 foods-11-04067-f006:**
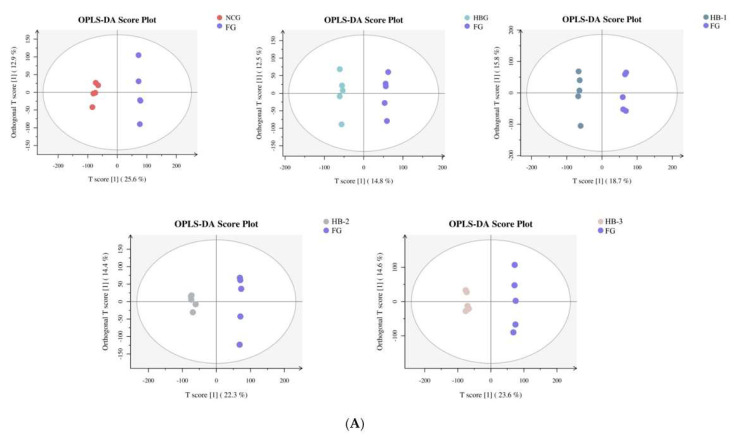
Orthogonal partial least squares discriminant analysis (OPLS-DA) score plots of the metabolome was visualized as follows: OPLS-DA in NCG vs. FG, HBG vs. FG, HB-1 vs. FG, HB-2 vs. FG, HB-3 vs. FG in ESI+ (**A**), and in ESI- (**B**).

**Figure 7 foods-11-04067-f007:**
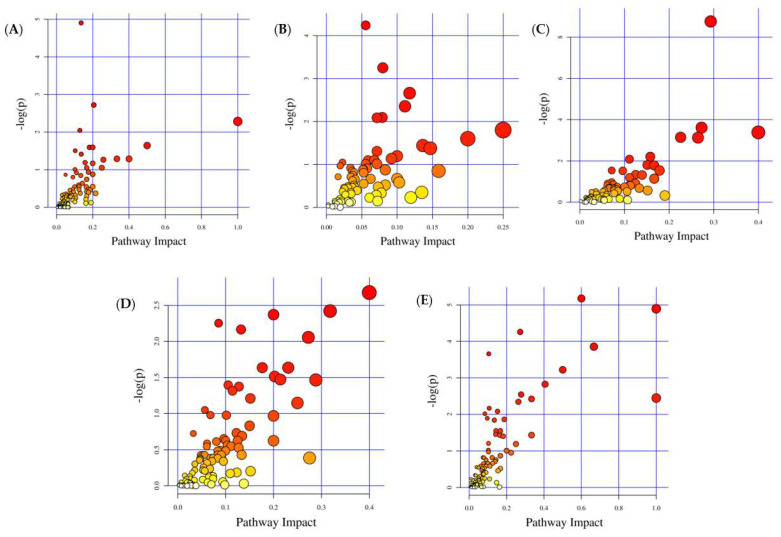
Different pathways analyses are shown as follows: (**A**) NCG vs. FG; (**B**) HBG vs. FG; (**C**) HB-1 vs. FG; (**D**) HB-2 vs. FG; (**E**) HB-3 vs. FG.

**Figure 8 foods-11-04067-f008:**
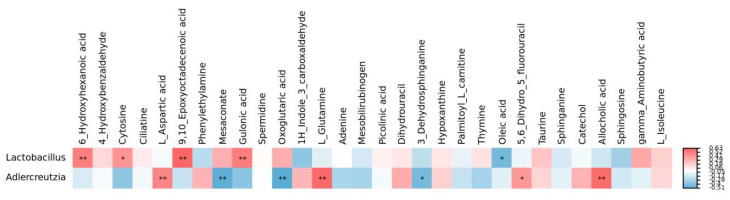
The related corrections between gut microbial composition based on genus level and significantly differential metabolites. * indicates a significant difference at *p* < 0.05; ** indicates a significant difference at *p* < 0.01.

**Table 1 foods-11-04067-t001:** Composition assessment of highland barley (HB), microwave fluidized highland barley (HB-I), extruded and puffed highland barley (HB-II), and ultrafine pulverized highland barley (HB-III).

Name	Method	Standard
β-glucan	Ultraviolet spectrophotometry method	Association of official analytical chemists, official methods of analysis (18th ed.)
Total polysaccharides	Phenol–sulfate method	DB12/T 847-2018
Dietary fibers	Enzyme hydrolysis	GB 5009.88-2014
Total anthocyanins	pH-differential method	[14]
Free phenols	Colorimetric method	[15]
Bound phenols	Colorimetric method	[15]
Moisture	Direct drying method	GB 5009.3-2016
Ash	Direct ignition method	GB 5009.4-2016
Protein	Kjeldahl method	GB 5009.5-2016
Fat	Soxhlet extraction method	GB 5009.6-2016
Total starch	Acid hydrolyzation	GB 5009.9-2016

**Table 2 foods-11-04067-t002:** Basic diet composition and nutrition levels for the studied mice.

Product	NCG		FG		HBG		HB-1		HB-2		HB-3	
	gm% 19	kcal% 20	gm% 22.62	kcal% 19.97	gm% 22.62	kcal% 19.97	gm% 22.62	kcal% 19.97	gm% 22.62	kcal% 19.97	gm% 22.62	kcal% 19.97
Protein												
Carbohydrate	67	70	45.51	40.18	45.51	40.18	45.51	40.18	45.51	40.18	45.51	40.18
Fat	4	10	20.06	39.85	20.06	39.85	20.06	39.85	20.06	39.85	20.06	39.85
Total	/	100	/	100	/	100	/	100	/	100	/	100
kcal/gm	3.85	/	4.53	/	4.53	/	4.53	/	4.53	/	4.53	
Ingredient	gm	kcal	gm	kcal	gm	kcal	gm	kcal	gm	kcal	gm	kcal
Casein, 80 Mesh	200	800	200	800	169	676	148	592	120	480	87	348
_L_-Cystine	3	12	3	12	3	12	3	12	3	12	3	12
Corn Starch	386.15	1544.6	212	848	0	0	0	0	0	0	0	0
Maltodextrin	125	500	71	284	0	0	0	0	0	0	0	0
Sucrose	200	800	124.41	497.64	124.41	497.64	124.41	497.64	124.41	497.64	124.41	497.64
Sample Fat	0	0	0	0	6	54	11	99	12	108	28	252
Sample Protein	0	0	0	0	31	124	52	208	80	320	113	452
Sample starch	0	0	0	0	283	1132	283	1132	283	1132	283	1132
Cellulose	50	0	50	0	50	0	50	0	50	0	50	0
Soybean Oil	25	225	25	225	19	171	14	126	13	117	0	0
Lard	20	180	155	1395	155	1395	155	1395	155	1395	152	1368
Mineral Mix S10020	5	0	5	0	5	0	5	0	5	0	5	0
Calcium Phosphate	13	0	13	0	13	0	13	0	13	0	13	0
Calcium Carbonate	5.5	0	5.5	0	5.5	0	5.5	0	5.5	0	5.5	0
Potassium Citrate,1 H_2_O	16.5	0	16.5	0	16.5	0	16.5	0	16.5	0	16.5	0
Sodium Chloride	2.59	0	2.59	0	2.59	0	2.59	0	2.59	0	2.59	0
Vitamin Mix V10001C	1	4	1	4	1	4	1	4	1	4	1	4
Choline Bitartrate	2	0	2	0	2	0	2	0	2	0	2	0
Cholesterol	0	0	11.25	0	11.25	0	11.25	0	11.25	0	11.25	0
FD&C Yellow Dye # 5	0	0	0	0	0	0	0	0	0	0	0	0
FD&C Red Dye # 40	0	0	0.05	0	0.05	0	0.05	0	0.05	0	0.05	0
FD&C Blue Dye # 1	0.1	0	0.05	0	0.05	0	0.05	0	0.05	0	0.05	0
Total	1054.84	4065.60	897.35	4065.64	897.35	4065.64	897.35	4065.64	897.35	4065.64	897.35	4065.64

Normal control group (NCG, fed with AIN-93M diet), high fat and cholesterol diet group (FG, XT108C diet), highland barley powder group (HBG, XT108C HFCD mixed highland barley), microwave fluidized highland barley HFCD group (HB-1, fed with XT108C mixed HB-1 diet), extruded and puffed highland barley HFCD group (HB-2, fed with XT108C mixed HB-2 diet), and ultrafine pulverized highland barley HFCD group (HB-3, fed with XT108C mixed HB-3 diet).

**Table 3 foods-11-04067-t003:** The BW, BWG, and FI of NCG, FG, HB, HB-1, HB-2, and HB-3.

	Week	NCG	FG	HBG	HB-1	HB-2	HB-3	*p*-Value
Body Weight (BW, gr)	1	38.33 ± 2.45	38.29 ± 5.64	39.27 ± 3.21	38.47 ± 2.11	39.81 ± 1.23	39.57 ± 3.11	
	2	39.37 ± 4.32	42.91 ± 3.21	42.14 ± 2.33	40.07 ± 3.44	42.24 ± 3.21	40.81 ± 2.11	
	3	39.68 ± 1.11	45.13 ± 1.23	46.55 ± 2.66	41.83 ± 5.55	43.50 ± 4.55	43.66 ± 3.12	
	4	40.04 ± 2.22	47.90 ± 3.22	50.84 ± 1.11	42.89 ± 6.66	44.08 ± 1.11	46.02 ± 2.12	
	5	40.52 ± 3.45	49.07 ± 1.22	51.77 ± 1.33	44.45 ± 7.44	47.78 ± 1.12	47.27 ± 1.24	
	6	41.26 ± 2.43	53.07 ± 5.67	52.55 ± 1.33	46.96 ± 3.21	48.86 ± 2.12	49.34 ± 1.44	
	7	41.97 ± 3.56	55.87 ± 4.32	53.9 ± 1.55	48.21 ± 2.45	50.24 ± 3.00	51.85 ± 5.13	
	8	42.26 ± 4.33	58.19 ± 0.97	57.49 ± 1.20	49.65 ± 2.78	52.30 ± 4.52	53.38 ± 3.12	
	9	43.89 ± 3.1	60.36 ± 4.30	58.50 ± 3.30	51.05 ± 4.65	53.77± 3.21	54.42 ± 3.21	
	10	45.11 ± 1.98	62.59 ± 2.11	61.00 ± 5.40	53.37 ± 3.65	54.50 ± 5.62	57.63 ± 1.22	
	11	47.45 ± 0.43	64.64 ± 5.40	63.2 ± 3.40	56.59 ± 4.13	57.01 ± 2.11	60.28 ± 4.55	
BWG		9.12 ± 2.12 ^e^	26.35 ± 3.20 ^a^	23.93 ± 3.21 ^b^	18.12 ± 2.13 ^d^	17.20 ± 1.34 ^d^	20.71 ± 1.21 ^c^	0.034
FI		5.41 ± 0.45 ^a^	4.16 ± 0.32 ^b^	4.13 ± 0.12 ^b^	4.11 ± 0.34 ^b^	4.14 ± 0.11 ^b^	4.11 ± 0.45 ^b^	0.029

Normal control group (NCG, fed with AIN-93M diet), HFCD group (FG, fed with XT108C diet), HB HFCD group (HBG, fed with XT108C mixed HB diet), HB-1 HFCD group (HB-1, fed with XT108C mixed HB-1 diet), HB-2 HFCD group (HB-2, fed with XT108C mixed HB-2 diet), HB-3 HFCD group (HB-3, fed with XT108C mixed HB-3 diet). Body weight = BW, body weight gain = BWG, feed intake = FI, and high fat and cholesterol diet. All data shown as mean ± SD. Values with different letters are significantly different at *p* < 0.05.

**Table 4 foods-11-04067-t004:** Effect of HB, HB-1, HB-2, and HB-3 on the serum biochemical parameters in HFCD-fed mice.

	NCG	FG	HBG	HB-1	HB-2	HB-3	*p*-Value
TG	0.91 ± 0.14 ^d^	1.79 ± 0.43 ^a^	1.42 ± 0.30 ^abc^	1.16 ± 0.04 ^bcd^	1.46 ± 0.02 ^ab^	1.01 ± 0.17 ^cd^	0.032
TC	3.67 ± 0.27 ^d^	7.637 ± 0.61 ^a^	6.59 ± 0.42 ^b^	5.14 ± 0.17 ^c^	6.66 ± 0.24 ^b^	5.72 ± 0.29 ^c^	0.021
HDL-C	6.35 ± 0.22 ^a^	4.31 ± 0.09 ^c^	5.32 ± 0.12 ^b^	3.63 ± 0.14 ^d^	4.99 ± 0.29 ^b^	4.47 ± 0.21 ^c^	0.031
LDL-C	0.47 ± 0.03 ^d^	1.68 ± 0.25 ^a^	0.76 ± 0.02 ^c^	0.75 ± 0.03 ^c^	0.66 ± 0.01 ^cd^	1.13 ± 0.21 ^b^	0.022
ALT	75.40 ± 3.26 ^c^	110.99 ± 5.39 ^a^	76.38 ± 2.74 ^c^	80.74 ± 0.81 ^c^	75.21 ± 3.21 ^c^	94.23 ± 3.87 ^b^	0.032
AST	138.71 ± 7.58 ^d^	246.22 ± 28.06 ^ab^	195.37 ± 4.88 ^c^	108.01 ± 1.80 ^e^	268.27 ±12.42 ^a^	228.27 ± 12.74 ^b^	0.025

Normal control group (NCG, fed with AIN-93M diet), HFCD group (FG, fed with XT108C diet), HB HFCD group (HBG, fed with XT108C mixed HB diet), HB-1 HFCD group (HB-1, fed with XT108C mixed HB-1 diet), HB-2 HFCD group (HB-2, fed with XT108C mixed HB-2 diet), HB-3 HFCD group (HB-3, fed with XT108C mixed HB-3 diet). All data shown as means ± SD. Values with different letters are significantly different at *p* < 0.05.

**Table 5 foods-11-04067-t005:** Statistical data of important differential metabolites in metabolomics.

Groups	Name	VIP	Fold Change	log2(FC)	*p*-Value	FDR
NCG vs. FG	Deoxycholic acid	1.562	78,755,000	26.231	0.007	0.122
	Myclobutanil	1.860	162,580	17.311	0.007	0.092
	4-Hydroxyestradiol	1.469	518.540	9.018	0.037	0.156
	Dicyclomine	1.486	82.590	6.368	0.012	0.092
	2-Phenylacetamide	1.760	60.707	5.924	0.012	0.122
	2-Methoxyestrone	1.443	59.446	5.894	0.012	0.092
	11-Dehydro-thromboxane B2	1.464	53.880	5.752	0.012	0.122
	Clomipramine	1.851	51.619	5.690	0.012	0.092
	Juvenile hormone III acid	1.881	49.980	5.643	0.012	0.092
	N1,N8-Bis(4-coumaroyl)spermidine	1.632	33.495	5.066	0.012	0.092
HBG vs. FG	Deoxycholic acid	1.131	58,735,000	25.808	0.007	0.311
	Myclobutanil	1.407	1,837,300	20.809	0.007	0.375
	3-Hydroxyflavone	1.743	66.308	6.051	0.012	0.311
	Secoisolariciresinol	1.816	13.186	3.721	0.037	0.399
	1,7-Dimethyluric acid	1.772	13.185	3.721	0.012	0.375
	Melibiitol	1.382	11.373	3.508	0.012	0.311
	1,2-Bis-O-sinapoyl-beta-D-glucose	2.001	11.184	3.483	0.012	0.311
	trans-Zeatin-7-beta-D-glucoside	1.974	10.722	3.423	0.022	0.419
	Retinoyl b-glucuronide	1.943	9.608	3.264	0.022	0.419
	Cellopentaose	1.909	9.392	3.231	0.012	0.311
HB-1 vs. FG	Deoxycholic acid	1.567	6,289,600,000	32.550	0.007	0.195
	Myclobutanil	2.263	157,820	17.268	0.007	0.165
	3-Epiecdysone	1.466	103.160	6.689	0.012	0.165
	2-Phenylacetamide	1.350	48.642	5.604	0.022	0.229
	Fluvoxamine	1.786	38.425	5.264	0.012	0.165
	Dicyclomine	1.937	34.795	5.121	0.012	0.165
	Chenodeoxycholic acid	2.175	23.294	4.542	0.012	0.195
	Sotalol	2.096	21.958	4.457	0.012	0.195
	N(6)-Methyllysine	2.200	18.581	4.216	0.012	0.165
	Pseudouridine	1.355	17.791	4.153	0.022	0.229
HB-2 vs. FG	Myclobutanil	1.673	10,872,000	23.374	0.007	0.134
	3-Hydroxyflavone	1.743	70.949	6.149	0.012	0.150
	4-Hydroxyproline	1.902	35.697	5.158	0.012	0.134
	4-Hydroxyestradiol	1.068	27.270	4.769	0.012	0.134
	Hydrogen phosphate	2.084	20.279	4.342	0.012	0.134
	LY 294002	1.332	13.496	3.755	0.022	0.160
	3,4-Dihydroxybenzeneacetic acid	1.374	13.489	3.754	0.037	0.212
	13S-hydroxyoctadecadienoic acid	1.783	12.782	3.676	0.012	0.134
	Isopropylparaben	1.254	12.155	3.604	0.012	0.150
	scyllo-Inositol	1.615	10.990	3.458	0.012	0.150
HB-3 vs. FG	Deoxycholic acid	1.299	2,480,000,000	31.207	0.007	0.162
	Myclobutanil	1.880	666,500	19.346	0.007	0.109
	3,4-Dihydroxybenzeneacetic acid	1.697	262.320	8.035	0.012	0.162
	2-Phenylacetamide	1.945	112.430	6.813	0.012	0.162
	Lipoxin B4	1.841	68.537	6.099	0.012	0.162
	N1,N8-Bis(4-coumaroyl)spermidine	1.713	43.511	5.443	0.012	0.109
	Homogentisate	1.937	41.340	5.370	0.012	0.162
	5-Dehydroavenasterol	1.598	28.466	4.831	0.012	0.109
	5-KETE	2.091	26.182	4.711	0.012	0.162
	11-Dehydro-thromboxane B2	2.225	22.065	4.464	0.012	0.162

Normal control group (NCG, fed with AIN-93M diet), HFCD group (FG, fed with XT108C diet), HB HFCD group (HBG, fed with XT108C mixed HB diet), HB-1 HFCD group (HB-1, fed with XT108C mixed HB-1 diet), HB-2 HFCD group (HB-2, fed with XT108C mixed HB-2 diet), HB-3 HFCD group (HB-3, fed with XT108C mixed HB-3 diet). FDR = false discovery rate. *p*-value: Statistically significant difference, *p*-value ≤ 0.05. VIP: OPLS-DA first principal component variable importance value projection, VIP ≥ 1. Fold change: Ploidy change between two groups, fold change ≥1.5.

**Table 6 foods-11-04067-t006:** Metabolic pathway analysis based on significantly differential metabolites.

Groups	Name	Total	Hits	Impact	Compounds
NCG vs. FG	Arachidonic acid metabolism	75	15	0.13613	C00427; C00584; C05950; C05961; C05964; C06315; C14717; C14732; C14748; C14749; C14773; C14774; C14775; C14794; C14810
	ABC transporters	138	13	0.094203	C00121; C00181; C00255; C00315; C00330; C00378; C00492; C00719; C00881; C01279; C01762; C03557; C05512
	Steroid hormone biosynthesis	99	11	0.1342	C00187; C00468; C00674; C00762; C03681; C03772; C05299; C05490; C05497; C05501; C13713
	Arginine and proline metabolism	78	9	0.11029	C00315; C00334; C00431; C02565; C03440; C03564; C03771; C04498; C10497
	Tryptophan metabolism	83	8	0.12387	C00954; C00978; C01717; C02693; C02937; C05635; C08313; C10164
	Serotonergic synapse	42	7	0.12903	C00427; C00584; C05635; C05964; C14773; C14774; C14775
	Glycine, serine and threonine metabolism	50	7	0.13665	C00097; C00109; C00258; C00263; C00719; C01005; C06231
	Purine metabolism	95	7	0.1016	C00147; C00330; C00385; C00802; C01762; C04051; C05512
	Cysteine and methionine metabolism	63	6	0.18538	C00097; C00109; C00263; C00606; C01005; C03145
	Caffeine metabolism	22	5	0.20588	C00385; C01762; C07130; C16361; C16362
HBG vs. FG	ABC transporters	138	11	0.07971	C00137; C00159; C00185; C00212; C00245; C00492; C00719; C01083; C01762; C01835; C03619
	Galactose metabolism	46	6	0.055556	C00137; C00159; C00492; C01613; C01697; C05399
	Steroid hormone biosynthesis	99	5	0.038961	C00674; C02140; C02538; C05497; C18040
	Phenylalanine metabolism	60	4	0.071429	C00042; C00601; C00642; C11457
	Pyrimidine metabolism	65	4	0.092079	C00106; C00337; C02170; C21028
	Biosynthesis of unsaturated fatty acids	74	4	0.052083	C06427; C08323; C16525; C16526
	Bile secretion	97	4	0.036697	C00504; C02538; C04483; C05466
	Caffeine metabolism	22	3	0.11765	C00385; C01762; C16356
	cAMP signaling pathway	25	3	0.11111	C00042; C00212; C01089
	Linoleic acid metabolism	28	3	0.078947	C14825; C14828; C14829
HB-1 vs. FG	Arachidonic acid metabolism	75	14	0.29319	C00219; C00427; C05951; C05957; C05964; C06315; C14717; C14732; C14748; C14749; C14774; C14794; C14807; C14810
	ABC transporters	138	9	0.065217	C00245; C00255; C00299; C00315; C00378; C00719; C00881; C01279; C05512
	Serotonergic synapse	42	6	0.22581	C00219; C00427; C05635; C05957; C05964; C14774
	Bile secretion	97	6	0.055046	C00315; C00695; C02528; C04483; C05122; C05466
	Primary bile acid biosynthesis	47	5	0.15122	C00245; C00695; C02528; C05122; C05466
	Neuroactive ligand-receptor interaction	52	5	0.096154	C00245; C00388; C00788; C05951; C13856
	Tryptophan metabolism	83	5	0.10574	C00978; C01717; C05635; C05831; C10164
	Taurine and hypotaurine metabolism	22	4	0.26471	C00227; C00245; C05122; C05844
	Lysine degradation	50	4	0.071429	C00431; C00449; C04020; C05161
	Steroid biosynthesis	58	4	0.067164	C01189; C01789; C05441; C05442
HB-2 vs.HB-2 vs.FG	ABC transporters	138	14	0.10145	C00009; C00059; C00064; C00121; C00123; C00135; C00140; C00243; C00299; C00315; C00719; C01157; C01762; C05349
	Tyrosine metabolism	78	11	0.2	C00042; C00146; C00355; C01161; C01384; C03765; C04043; C05576; C05580; C05582; C06199
	Tryptophan metabolism	83	10	0.20242	C00108; C00632; C00780; C00955; C01598; C02693; C05643; C05831; C08313; C10164
	Nicotinate and nicotinamide metabolism	55	7	0.28837	C00042; C00153; C00253; C01384; C03150; C05380; C15987
	Steroid hormone biosynthesis	99	7	0.056277	C00523; C02140; C05301; C05471; C05497; C05501; C18040
	Central carbon metabolism in cancer	37	6	0.13208	C00042; C00064; C00123; C00135; C00152; C00158
	Protein digestion and absorption	47	6	0.12766	C00064; C00123; C00135; C00146; C00152; C18319
	Arachidonic acid metabolism	75	6	0.04712	C00639; C00696; C04853; C05957; C06315; C14717
	Arginine and proline metabolism	78	6	0.061275	C00315; C00431; C01157; C02565; C03296; C03771
	Alanine, aspartate and glutamate metabolism	28	5	0.085227	C00042; C00064; C00152; C00158; C12270
HB-3 vs.FG	Arachidonic acid metabolism	75	12	0.10471	C00584; C02165; C05949; C05952; C05961; C05964; C06315; C14732; C14748; C14769; C14774; C14775
	ABC transporters	138	11	0.07971	C00009; C00059; C00140; C00212; C00315; C00492; C00881; C01157; C01279; C01762; C05349
	Tyrosine metabolism	78	10	0.084	C00146; C00232; C00544; C00642; C01161; C01384; C03765; C05594; C06199; C10447
	Steroid hormone biosynthesis	99	9	0.13853	C00187; C00523; C00535; C01176; C02140; C05138; C05301; C05471; C05490
	Neuroactive ligand-receptor interaction	52	7	0.13462	C00212; C00388; C00584; C01516; C01598; C02165; C05952
	Steroid biosynthesis	58	7	0.14179	C00187; C01189; C01673; C01789; C05441; C15783; C15808
	Arginine and proline metabolism	78	7	0.07598	C00315; C00431; C01157; C02565; C02946; C03296; C03564
	Tryptophan metabolism	83	7	0.081571	C00632; C00954; C01598; C02693; C05643; C05831; C10164
	Ovarian steroidogenesis	24	6	0.27273	C00187; C00535; C01176; C05138; C05301; C14769
	Serotonergic synapse	42	6	0.096774	C00584; C02165; C05964; C14769; C14774; C14775

Total, the total number of metabolites in the target metabolic pathway; Hits, the number of differential metabolites in the target metabolic pathway, Raw *p*, the *p* value of the hypergeometric distribution test; Impact, metabolic pathway impact value; compounds, metabolite KEGG ID.

## Data Availability

The data presented in this study are available on request from the corresponding author.

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
