# Peer review of "Modified Highland Barley Regulates Lipid Metabolism and Liver Injury in High Fat and Cholesterol Diet ICR Mice"

_foods, 2022, doi:10.3390/foods11244067_

Round 1
Reviewer 1 Report
Appreciable authors
In the included pdf file called "foods-2013572-peer-review-v1jjvr.pdf" you will find the suggestions and corrections of the review, please consult that file.
These are the most relevant suggestions (in the pdf there are more details and other corrections, take into account all the indications)
1) The description of the experimental design is incomplete or lacks more information on how it was established. Were the mice identified individually? How were they identified? How were the experimental groups formed? randomly? Specify this in the methodology. Also, how much food was fed? how often? How was the food fed? Were the experimental groups fed the identical diet? Or was there variation in the proportion of the components? Specify. What percentage of AIN-93M/HG, or of XT108C/HG was used in the different mixtures? Provide these data. In addition, it is missing to provide the most relevant data or information of the diets called AIN-93M and XT108C.
2) It is suggested to use a simpler abbreviation system for the experimental groups than those used. There is even a duplicate in the name of some groups, such as the experimental groups called HB-1, HB-3 and HB-2 in Table 2 are confused with the modified Highland barley described in paragraph 90-92 (HBM), also called HB-1 (microwave fluidized HB), HB-2 (extruded and puffed HB), HB-3 (ultrafine pulverized HB). Also, the table itself describes the FG and HBG groups, of which the meaning of these abbreviations is not specified. Then, for the clarity of the text, a simple and clear nomenclature system for the experimental groups should be ordered and searched for, or if it is desired to continue using these abbreviations, they should be clearly described and specified. In the revised pdf there are more observations and suggestions regarding this, please review them.
3) In the same sense as point 2), the authors use many abbreviations (some of which their meaning has not even been specified) which makes it difficult to read the manuscript and remember their meaning. It is sometimes suggested to place the meaning of some abbreviations in parentheses when referring to them.
4) Support some experimental methods with techniques or standards (see revised pdf)
5) Arrange the tables, standardizing the number of decimals in the data, incorporating the units where necessary, and choosing a font size that prevents the signs from appearing cut off (see revised pdf). Apply the changes following the journal's author guide for tables.
6) For all figures Improve the quality of the images since they are blurred and increase the size of the signs and numbers on the axes, because the titles and numerical scales cannot be read
7) Bibliographically compare the results obtained (paragraphs 282-310 and 321-350, for example) with the results of similar published research. Also bibliographically support some paragraphs, such as 351-352 and 354-357, for example (see revised pdf).
I invite you, if possible, to dispense with some figures if this is possible to reduce the size of the document and improve the writing, which would improve the quality of the manuscript. Thank you

Author Response
Thank you very much for your professional comments. Please see the attachment.

Reviewer 2 Report
Interesting work and study which is presented by the authors and it adds more details in the application of β-glucan as an active ingredient which interests the producers/suppliers as an active ingredient in natural health product. More human clinical studies are required in order to enforce the regulatory claims for this ingredient. However, the main concern is about the allergenicity of oat/barely/wheat derived ingredients. How authors can address this issue?
The manuscript included large number of data. If not significant, part of data reports can be submitted from manuscript as supplementary information/data.
Please be focus and highlight the significant outcomes relevant to conclusion.
Minor observations:
Title: Please minimize the length of the title.
Line 453: Some phrasal/grammatical issue to be revised.
Quality of graphs in Fig 4 and 5 can preferably increased.
Author Response

(The authors gave the same response as above.)

Reviewer 3 Report
· The title is too long and difficult to read. It needs to be simple, clear, specific and reflects the content of the manuscript.
· A brief background to understand the context for the study is needed at the beginning of abstract.
· Please provide details about type of analysis used and significant p-value in abstract, so that results can be more easily understood and interpreted.
· Line 11-14: The study design and the selected groups should be clearly defined.
· Introduction: The novelty aspect is particularly weak (Line 82-87). Authors provided few references to support the aim, with some of these are addressed thoroughly. Are they in vivo or in vitro study? It would benefits if the authors comprehended and compared the in vitro studies or the results of clinical trials (there are many) with dietary intervention supplements (Line 41-45). In general, please refine and rephrase the whole introduction using more studies.
· Line 93-99: This section is vague. Please describe in details how HB-1, HB, HB-3 and HB-2 were assessed?
· Line 189-196: I think this information should be to statistical analysis section.
· Table 3 & 4: Please include a column that shows significant P-value. It is unclear what types of statistical tests were used in these tables?
· All figures should be a reasonable size and have a high resolution for the data to be clear.
· A discussion should be prepared by organizing information according to: the main findings and comparison of these findings with those reported in the literature; the hypothesis about the non-significantly differences, the strengths and weaknesses of the study and in relation to other studies, information about the present analyses and the implications of this study and future research directions. A lot of statements were reported without discussed; e.g., how amino acids play a significant role in ameliorating hyperlipemia and reducing inflammation? (Line 453-469). How SCFA as an epigenetic modifier could reduce obesity? Please refer to these interesting articles (Front Genet. 2019; 10: 1329; Nutrients. 2021 Oct 21;13(11):3702; Adv Nutr. 2019 Jan; 10(Suppl 1): S17–S30).
· The manuscript requires significant editing by a native English speaker.
Author Response

(The authors gave the same response as above.)

Round 2
Reviewer 1 Report
I believe that most of the suggestions made were applied. A form correction is suggested in the 2nd paragraph of page 4, as can be seen in the attached revised file.

Reviewer 3 Report
No further comments.